# Characterization of acute effects of football competition on hamstring muscles by muscle functional MRI techniques

Sandra Mechó [1,2]*, Alicia Palomar-Garcia[3]*, Manuel Wong[2], Juan C. Gallego[1,2], Francesc López[1,2], Xavier Valle[2], Ferran Ruperez[2], Ricard Pruna[2], Juan R. González[4,5,6], Gil Rodas [2,7,8]

1 Department of Radiology Hospital de Barcelona, SCIAS, Barcelona, Spain, 2 Medical Department of Futbol Club Barcelona (FIFA Medical Center of Excellence) and Barça Innovation Hub, Barcelona, Spain, 3 Canon Medical Systems Spain and Portugal, Barcelona, Spain, 4 Barcelona Institute for Global Health (ISGlobal), Barcelona, Spain, 5 Universitat Pompeu Fabra (UPF), Barcelona, Spain, 6 Centro de Investigación Biomédica en Red en Epidemiología y Salud Pública (CIBERESP), Barcelona, Spain, 7 Sports Medicine Unit, Hospital Clinic and Sant Joan de Déu, Barcelona, Spain, 8 Faculty of Medicine and Health Sciences, University of Barcelona, Barcelona, Spain

* mechomeca@gmail.com (SM); alicia.palomar@eu.medical.canon (APG)

**Data Availability Statement:** All relevant data are within the manuscript.

## Abstract

Muscle functional MRI identifies changes in metabolic activity in each muscle and provides a quantitative index of muscle activation and damage. No previous studies have analyzed the hamstrings activation over a football match. This study aimed at detecting different patterns of hamstring muscles activation after a football game, and to examine inter- and intramuscular differences (proximal-middle-distal) in hamstring muscles activation using transverse relaxation time (T2)–weighted magnetic resonance images. Eleven healthy football players were recruited for this study. T2 relaxation time mapping-MRI was performed before (2 hours) and immediately after a match (on average 13 min). The T2 values of each hamstring muscle at the distal, middle, and proximal portions were measured. The primary outcome measure was the increase in T2 relaxation time value after a match. Linear mixed models were used to detect differences pre and postmatch. MRI examination showed that there was no obvious abnormality in the shape and the conventional T2 weighted signal of the hamstring muscles after a match. On the other hand, muscle functional MRI T2 analysis revealed that T2 relaxation time significantly increased at distal and middle portions of the semitendinosus muscle (p = 0.0003 in both cases). By employing T2 relaxation time mapping, we have identified alterations within the hamstring muscles being the semitendinosus as the most engaged muscle, particularly within its middle and distal thirds. This investigation underscores the utility of T2 relaxation time mapping in evaluating muscle activation patterns during football matches, facilitating the detection of anomalous activation patterns that may warrant injury reduction interventions.

**Funding:** The author(s) received no specific funding for this work.

**Competing interests:** The authors have declared that no competing interests exist.

## Introduction

Muscle injuries are the most common category of injuries in athletes, accounting for more than 30% of injuries in football players [1–3]. Specifically, hamstring muscles injury has been reported to be the most common injury subtype in women and men's elite-level football players, constituting 12–24% of all time-loss injuries. Around 18% of all reported hamstring injuries were recurrent with over two-thirds occurring within 2 months after footballer's return to play [4, 5].

The most common mechanism of injury to the hamstrings is high-speed running [6–10]. Based on previous research on hamstring lengthening and internal forces during normal walking and running, maximum hamstring muscle tension during high-speed running occurs at the end of the swing phase being the muscular damage normally at the proximal myotendinous junction of biceps femoris and semitendinosus muscles [7, 11–13]. Therefore, the most common mechanism of injury to the hamstrings is known, but not all players get injured and it is not always at the same moment in the season, match or training session, so it is not the only cause for the injury, but it is multifactorial. Intrinsic and extrinsic risk factors to the player exist [14]. Intrinsic risk factors may include: residual weakness after previous hamstring injury, strength asymmetry of hamstrings, lack of eccentric strength of the hamstrings, fatigue, poor core stability, poor coordination, player wellness (sleep patterns, relationships, etc), poor flexibility and, poor nutrition [13, 14]. It is possible that within the intrinsic factors to the player there may be different patterns of muscle activation or recruitment during a football match. Further knowledge of the relationship between muscle activation or recruitment patterns and the mechanical properties of the set of movements performed by a football player during a match should allow a better understanding of the characteristics of each individual or position on the field. In addition, it could help assess the risk of suffering muscle injuries.

In order to detect different recruitment patterns it is necessary to obtain correct measurements of spatial type and intensity of muscle activation [15]. The standard assessment of muscle recruitment during exercise includes the use of surface electromyography, muscle force measurement, and muscle needle biopsy technique (glycogen depletion, aerobic and anaerobic enzyme activities) [16, 17]. Furthermore, muscle functional magnetic resonance imaging (mfMRI) is another method of assessing muscle recruitment, of increasing importance, which reflects the biochemical changes that occur in activated muscle fibers as changes in T2 relaxation time that can be measured [17–22]. Each tissue has its own T2 relaxation time which can change according to different physiological and chemical conditions. Activated muscles usually appear hyperintense on T2-weighted sequences, so they present longer T2 relaxation times. Some studies indicate that there is a linear relationship between the increase in T2 relaxation time and the intensity of the exercise previously practiced [19, 23].

Evaluating the muscle groups that are activated during a football match, depending on the individual characteristics of the players or the position on the field, could provide greater insight into the specific muscle recruitment patterns that could be the target of a personalized training program for the prevention of muscle injuries and a more specific rehabilitation program.

Since the half-life of exercise-induced changes in muscle T2 has been shown to be approximately 7 minutes [18, 24], it is necessary to have the MR equipment as close as possible to the field where a training session or football match takes place. The Sports Medicine Department of FC Barcelona includes an MR Center, and this department is inside the Ciutat Esportiva of FC Barcelona where there are training sessions every day and football matches almost every weekend.

To date, no study has been carried out to assess muscle recruitment of the thighs in professional football players after a match of at least 60 minutes. To overcome this limitation, in this

study we set out to identify the sites of the hamstring muscles that are significantly activated during a football match using mfMRI. To this end, we measured football-match induced changes in T2 values in the proximal, middle and distal portions of each hamstring muscle (long head biceps femoris, short head biceps femoris, semimembranosus and semitendinosus).

## Materials and methods

### Participants

Eleven male football players without acute muscle lower extremity injuries were recruited for this study over season 2021–22 (from March 13, 2022 to December 18, 2022). All the participants were members of the FC Barcelona junior team (16–18 yr).

This study was approved by the local research ethics committee of Barça Innovation Hub and Ethics Committee for Clinical Research of the Catalan Sports Council with the reference number 10/2019/CEICEGC. The individual in one of the figures of this manuscript has given written informed consent (as outlined in PLOS consent form) to publish these case details.

The players and staff were informed about the purpose, procedures, and risks of the study, and written informed consent was obtained from each participant.

### Procedures

**Data collection.** ll participants played at least 60 minutes of an official football match. MRI scans were acquired before (2h) and immediately after (on average 13 min) a football match to avoid potential tissue recovery. Once the player stopped playing, he drunk an energetic drink to minimize cramps during the scan that could occur because of postexercise status. In order to prevent additional physical effort the player was transferred from the football pitch to the MR scanner in a golf cart and an MR compatible wheelchair. The transfer from the pitch to the MR facility and the preparation of the subject for the scanning session took about 7 minutes. This procedure can be represented by this collection of images (Fig 1).

GPS data were recorded in order to measure the total load of the game and ensure that all players obtained a level of total load similar to the average of all matches. GPS data were collected using the WIMU PROTM device (RealtrackSystems S.L., Almeria, Spain). Intra- and inter-unit reliability was acceptable (intra-class correlation coefficient value was 0.65 for the x-coordinate, and 0.85 for the y-coordinate) for the systems analyzed [25, 26]. The data collected were analyzed using the SPROTM Software (version 927; RealtrackSystems, Almeria Spain), which exports the data in RAW format.

The GPS variables that have been considered for this study are the following ones: total time in minutes; total distance in meters; high metabolic load distance (HMLD) in meters; high speed running (HSR) in meters (those at speeds greater than 21 km/h); maximum speed in km/h; number of accelerations higher than 3 m/s2 and number of decelerations higher than -3 m/s2.

**MRI acquisition.** Thigh MRI scans were performed on a Vantage Galan 3T system (Canon Medical Systems, Tochigi, Japan) using two overlapped 16-channel Atlas SPEEDER body coils. The participants were placed on the MR table in a supine and relaxed position with the legs tied together to reduce artifacts. Multi-echo 2D FSE T2-weighted imaging was acquired with the following parameters: TE 20, 60, 100 and 140 ms; TR 3200 ms; in-plane resolution 0.25 x 0.25 mm; slice thickness 3 mm; gap 15 mm and scan time 2:05 min. Pre-match acquisitions also included axial fat-suppressed proton density-weighted (PD FS) imaging with TE 33 ms; TR 2954 ms; in-plane resolution 0.49x0.49 mm; slice thickness 3 mm; gap 15 mm and scan time 1:50 min. The imaging protocol was conducted in two blocks to cover the extension of hamstring muscles at the proximal, middle and distal portions.

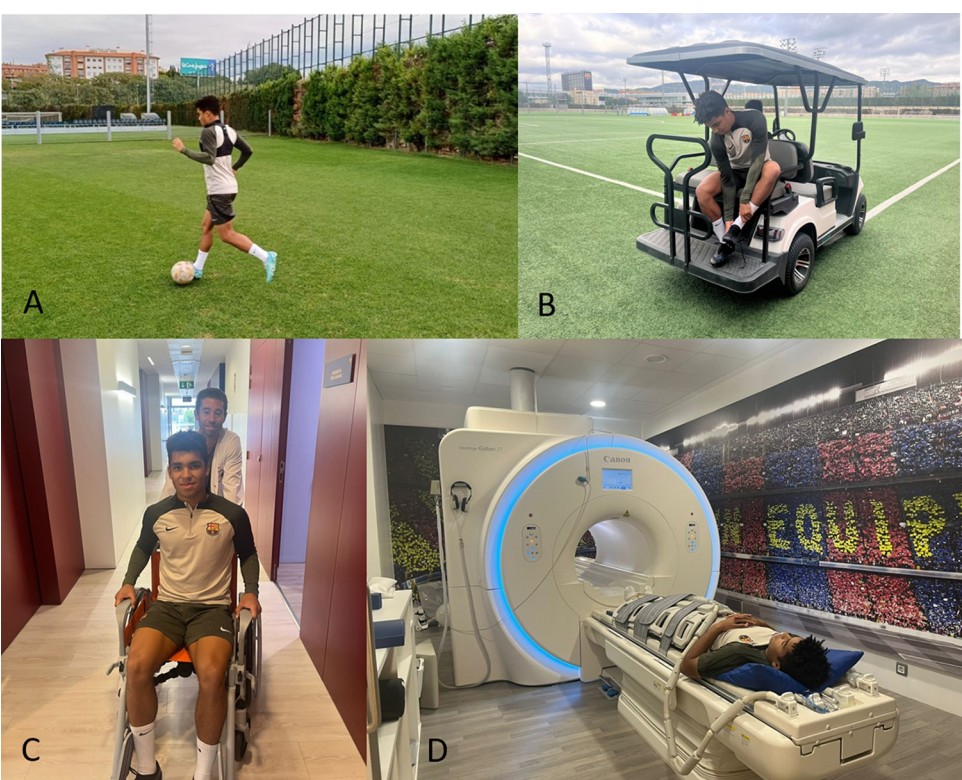

**Fig 1. Representation of the procedure between the field of play and the player placement in the MR scanner for data collection.** A) Player during the match, B) player on the golf cart to the sports medicine center of the Joan Gamper Sports City, C) player on the MR compatible wheelchair in MR facilities hallway, D) player in the MR scanner.

## Data analysis

The multi-echo 2DFSE T2-weighted images were processed with the Relaxometry plug-in in Olea Sphere® software (Olea Medical, La Ciotat, France) to generate the T2 relaxation time maps. Regions of interest (ROIs) were drawn manually on the different hamstring muscles by an experienced radiologist specialized in musculoskeletal imaging to obtain the mean T2 values for each portion of each muscle (three portions for each, but in case of short head of biceps femoris just two portions: proximal and distal). An average of 5 slices were discarded due to motion artifacts. Pre-match PD FS images were used for anatomical reference to avoid placing ROIs on areas with intramuscular fat, scars or blood vessels. For consistent comparison, ROIs of same size and shape were used to measure pre-match and post-match T2, as illustrated in Fig 2. In addition, to ensure signal homogeneity within the measurements, standard deviation was checked to be as low as possible (less than 6 ms). Mean difference in T2 values between pre-match and post-match scenarios was calculated to characterize T2 time changes resulting after physical activity.

## Statistical analyses

Different measurements at each ROI and each leg before and after a match were analyzed using a linear mixed-effect model incorporating a nested random effect of individual within leg. In total, we ran 11 different models corresponding to each ROI (Biceps Femoris Lh, Biceps Femoris Sh, Semimembranous and Semitendinous) and each measurement (distal, middle and proximal). The primary goal of this investigation was to assess differences in muscle

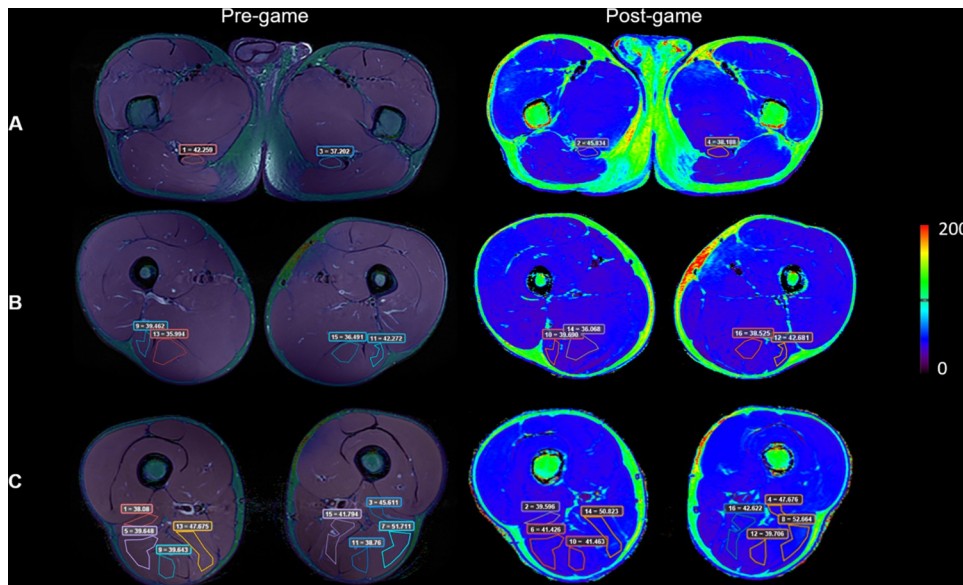

**Fig 2. Illustration of T2 measurement methodology on hamstrings muscles.** Pre-match overlay of PD FS images and T2 maps (left) and post-match T2 maps (right). A) ROIs on left and right proximal semitendinosus. B) ROIs on left and right proximal biceps femoris long head and middle semitendinosus. C) ROIs on left and right proximal biceps femoris short head, middle biceps femoris long head, middle semimembranosus and distal semitendinosus.

activation between the two legs before and after the match, while accounting for inter-individual variability. Hypothesis testing was conducted to ascertain the statistical significance of observed differences using the score test. Multiple correction was performed as necessary to avoid false positive results given by the fact we are using the same data to analyze 11 different model (Bonferroni corrected p-value: 0.004). In a second step, potential confounding variables, obtained from GPS data, were controlled for in the analysis. This mixed-effects modeling strategy facilitated an effective examination of muscle activation differences in the context of a football match, providing insights into the nuances of individual responses within and between legs.

## Results

### Participants

Eleven male football players were included in the study. Patient data are provided in Table 1.

**Table 1. Characteristics of all athletes.**

|  | Mean ± SD |  |
|---|---|---|
| **Age, years** | 17.5 ± 0.7 |  |
| **Height, cm** | 177.1 ± 8.2 |  |
| **Weight, kg** | 70.3 ± 5.2 |  |
| **Body mass index, kg/m$^2$** | 22.4 ± 1.77 |  |
| **Field position** | **Number** | **%** |
| Midfielder | 2 | 18.2 |
| Defending | 8 | 72.7 |
| Forward | 1 | 9.1 |

Data are reported as Mean and Standard Deviation (SD); and number and percentage (%) of athletes.

**Table 2. GPS device data collection.**

| GPS data | Mean ± SD |
|---|---|
| Total time, min | 82.7 ± 9.5 |
| Total distance, m | 8,783.4 ± 1,214.4 |
| HMLD, m | 1,680 ± 239.04 |
| HSR, m | 461.8 ± 146.9 |
| Max speed, km/h | 30.1 ± 1.6 |
| Acc>3m/s$^2$ | 56.7 ± 14.6 |
| Dec>-3m/s$^2$ | 61.4 ± 16.9 |

Data are reported as Mean and Standard Deviation (SD)

## Procedure

**Data collection.** The values from the GPS device are provided in Table 2.

## Data analysis

T2 mapping MRI-based data of the recruited participants are summarized in Table 3.

## Statistical analyses

No correlation was found between the different parameters of the GPS system and the magnitude of the change in T2 relaxation time.

After correcting for multiple comparisons, the analysis based on linear mixed models (illustrated in Fig 3) showed a significant bilateral activation of the semitendinosus muscle at middle (p = 0.0003) and distal (p = 0.0003) portions. When stratifying by leg, significant activation at middle portion of the semitendinosus muscle was found in the right leg (p = 0.0038), while results from the left leg showed no significant activation in any portion of any muscle.

## Discussion

We determined whether there are inter- and intramuscular differences in hamstring muscle activation after a football match using mfMRI. The main finding of the present study was that

**Table 3. T2 mapping MR-based data.**

| | | | Pre-match T2 (ms) | | | Post-match T2 (ms) | | |
|---|---|---|---|---|---|---|---|---|
| | | | Left | Right | Both | Left | Right | Both |
| **BFLh** | Proximal | | 39.20 ± 2.64 | 36.16 ± 1.55 | 37.68 ± 2.62 | 39.4 ± 2.31 | 38.15 ± 2.13 | 38.78 ± 2.26 |
| | Middle | | 43.49 ± 4.38 | 37.08 ± 1.52 | 40.29 ± 4.58 | 44.06 ± 4.69 | 38.40 ± 1.45 | 41.23 ± 4.46 |
| | Distal | | 41.81 ± 2.26 | 37.18 ± 1.16 | 39.50 ± 2.95 | 42.51 ± 2.36 | 38.39 ± 1.36 | 40.45 ± 2.83 |
| **BFSh** | Middle | | 45.52 ± 4.35 | 37.74 ± 1.26 | 41.63 ± 5.06 | 46.44 ± 3.34 | 38.64 ± 1.34 | 42.54 ± 4.70 |
| | Distal | | 42.97 ± 3.38 | 38.81 ± 1.69 | 39.89 ± 4.09 | 43.33 ± 3.95 | 37.61 ± 1.22 | 40.47 ± 4.09 |
| **ST** | Proximal | | 35.20 ± 0.90 | 39.39 ± 1.48 | 37.29 ± 2.45 | 36.11 ± 1.44 | 39.63 ± 2.15 | 37.87 ± 2.54 |
| | Middle | | 35.40 ± 1.05 | 35.86 ± 0.63 | 35.63 ± 0.87 | 35.98 ± 1.23 | 36.90 ± 1.19 | 36.44 ± 1.27 |
| | Distal | | 37.03 ± 1.72 | 38.39 ± 1.66 | 37.71 ± 1.79 | 38.25 ± 2.05 | 40.07 ± 2.38 | 39.16 ± 2.36 |
| **SM** | Proximal | | 36.61 ± 1.28 | 42.85 ± 2.68 | 39.73 ± 3.79 | 36.95 ± 1.78 | 42.71 ± 2.45 | 39.83 ± 3.61 |
| | Middle | | 38.04 ± 1.07 | 43.13 ± 2.13 | 40.58 ± 3.08 | 38.52 ± 1.73 | 43.68 ± 2.62 | 41.10 ± 3.42 |
| | Distal | | 37.80 ± 1.25 | 39.84 ± 1.86 | 38.82 ± 1.87 | 38.29 ± 1.14 | 39.79 ± 1.51 | 39.04 ± 1.52 |

Pre-match and post-match T2 miliseconds values (mean ± sd) of the hamstring muscles at different thigh portions. T2 values were reported for each leg and on average.
Abreviations: BFLh, biceps femoris long head; BFSh, biceps femoris short head; ST, semitendinous and SM, semimembranous.

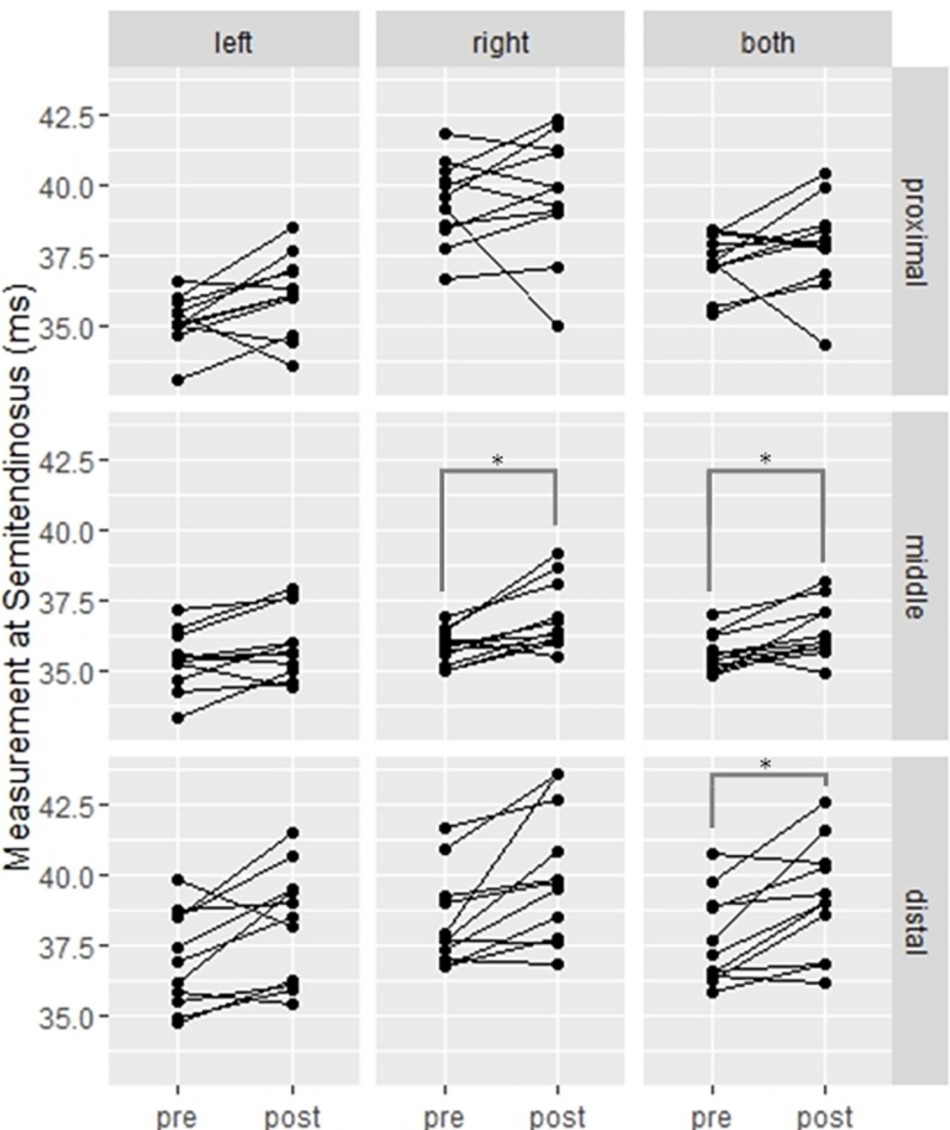

**Fig 3. T2 value difference between pre- and post-match of the semitendinosus muscle.** *Significantly different from baseline (pre) with p<0.004.

the semitendinosus muscle was the most activated hamstring muscle during the match exercise. No significant intermuscular differences were found in terms of T2 change between the other hamstring muscles.

In order to ensure that all players obtained a level of load similar to the average of all matches we analyzed the GPS data. Our sample was made up of youth players. It is known that there are no differences in GPS upload metrics between youth players and professionals [27]. In fact, as shown previously, HMLD, HSR, PL and TD were slightly higher in youth players, compared to the professional category [28]. In the Guitart et al study there were similar external GPS load metrics by playing position [26]. If we evaluate the HSR as a representative date of the intensity of the match, in the study by Guitart et al the average was 444.41 m, while our result was 461.1 m. Therefore, we derived the goal from the GPS data to ensure that the load level of the match was similar to the average of all matches. A statistically significant difference

in muscle activation could be detected due to the level of high-speed running during the game, with a maximum speed recorded by the players of 30.1km/h. Schache et al found that there was no statistical difference in the magnitude of maximal stretch between the hamstring muscles during slower speed (<18 km/h) [11].

At the beginning of the present study our hypothesis was that the greatest muscle activation would be found in the BFLh muscle, in line with a previous work [29]. However, the results did not support this. Our hypothesis was based on previous studies suggesting that peak muscle-tendon stretch is greater in the long head of the biceps femoris during running compared to the semitendinosus and semimembranosus [9, 13, 30]. However, our results are consistent with other works that reported that the semitendinosus presented the greatest muscle activity and was more recruited than the biceps femoris and the semimembranosus in strength exercises and in locomotion [9, 11]. This activation pattern seems to be the result of complex neuromuscular coordination within the hamstring muscle group, which arguably provides the most efficient muscle force production [9]. In fact, it has been considered that a relatively high activity of the semitendinosus may protect against injury, possibly because it would unload the most frequently affected long head of the biceps femoris [9]. In contrast to our findings, a previous work with basketball players using electromyography concluded that in the late-swing phase during high-speed running, the semimembranosus was the most active muscle [7]. As these results are different from those obtained so far [10, 12, 31], a possible explanation would be the heterogeneous distribution of activity within the individual hamstring muscles and the different electrode placement [7].

Furthermore, in our study, T2 significantly increased in the distal and middle portions of the semitendinosus muscle, although those values in the proximal portion remained constant. Akima et al indicated that the activation of subdivisions within a single muscle is heterogeneous relative to the task, and this subdivision of muscle compartments may act to control muscle activation originally modulated by the nervous system [17]. Our results are consistent with those obtained by Higashihara et al, who analyzed muscle activation before and after a marathon race [29]. They also found a significant increase in T2 times in the distal and middle portions of the biarticular hamstring muscles one and three days after the marathon. They considered these increases in the T2 values as a sign of inflammatory oedema that takes several days to recover, calling it muscle damage induced by prolonged running. We did not evaluate the increase in the T2 value days after the match, so we did not use the term muscle damage, but only muscle activation. A possible explanation for the greater activation in the middle and distal portions of the semitendinosus is that the location of muscle activation after a match, in a prolonged running, is influenced primarily by the intensity of the repetitive negative work performed by the lower extremity. Football belongs to the sports in which a high rate of hamstring strain injury has been reported [1], as it requires prolonged, high-intensity intermittent performance, and players cover long distances during a game at various speeds; in fact, movement distance with a higher-speed running represents only 5% of the total distance. In addition, it is a sport where players perform decelerations(>-3m/s2) and accelerations (>3m/s2) [32]. In our study, the average of decelerations was greater than the number of accelerations. Therefore, the negative work was greater. Silder et al also reported that during eccentric knee flexion, the greatest tissue movement in the BFlh muscle is observed along the distal myotendinous junction [33]. However, our results are inconsistent with the clinical observations that the proximal myotendinous junction of BFLh muscles is most commonly affected in acute hamstring muscle injury [1]. This discrepancy could be explained by the possible difference in the behaviour of the hamstrings in high-speed running and in prolonged running [29].

It has been reported that recruitment of the working muscles is dependent on type of contraction and velocity. Repetitive isokinetic knee extension exercise results in greater activation

of the rectus femoris than the rest of quadriceps muscles. However, isotonic knee extension exercise does not have the same results [17, 34–37]. Kuling et al demonstrated that the biceps brachii is more activated in fast eccentric contraction task (2 seconds) and the brachialis is more activated in slow contraction task (10 seconds) [38]. Fisher et al also demonstrated that there is a correlation between the generated force during exercise and contrast T2-weighted MRI enhancement [17, 24]. In the present study, no correlation was found between the different parameters of the GPS system and the magnitude of the change in T2 relaxation time. One possible reason is that there are several changes of pace and movements during a match, and the player is not active throughout the game, so the degree of repetitiveness and force generation is lower. Another hypothesis that could explain the lack of correlation with GPS parameters is that the post-match T2 mapping focuses mainly on the activity that took place at the end of the match, that is, in the last minutes. The circumstances of the match, i.e. winning or losing, dictate a lot about the pace and intensity of the activity. In all the cases evaluated in this study, the team to which the analyzed player belonged was winning and this may have caused a more defensive than offensive attitude. Regarding what was stated about the MRI mainly representing the last minutes of exercise, one of the inclusion criteria in the study was that at least the last 15 minutes had been played without rest. Therefore, the minimum limit of minutes played was 60, 45 minutes in the first half and 15 minutes in the second, with rest between them.

To our knowledge, this is the first study to describe this effect on the hamstrings in football players. A possible limitation of our study is the use of a fast spin echo sequence to measure T2 relaxation time, instead of a spin echo sequence, which has been reported as a more accurate technique [39]. However, for our purpose, it was crucial to reduce the scan time as much as possible to have a minimum period of time between the end of the match and the MRI acquisition. In this way, we could ensure that we characterized the acute effects of a football game on muscle activation. Furthermore, the aim of the study was to evaluate the variation in T2 relaxation time from baseline, so the precision of the estimation was not paramount because potential overestimation would be present at both time points, so subtraction would eliminate any bias. Another limitation indicated is that in all cases an average of 5 slices were discarded because the players presented postexercise cramps. But in no case did it involve loss of information from an entire portion of the hamstring tendon, so no case was ruled out due to movement artifacts.

We have not been able to evaluate different muscle recruitment patterns depending on individual players characteristics or their position on the field of play. Future studies should be conducted with larger cohorts of players to evaluate, for example, any potential effect that the player's position on the field could have on the muscle recruitment pattern.

Through this study we have been able to demonstrate that mfMRI is a good method to analyze the activation of the hamstring muscles in football players. It has been shown that the semitendinosus is the most activated hamstring muscle during a football match and specifically in its middle and distal portion. Based on our results, conditioning that focuses on the distal and middle portions of the hamstring muscles may be more useful in preventing injury after football match.

## Clinical implications

The information derived from this investigation should allow the physician, strength and conditioning coach or rehabilitation specialist a better understanding of the site-specific activation of the posterior thigh muscles during a football match. It is well known that the most injured muscle in football is the biceps femoris. The fact that we have seen that the semitendinosus is

the most activated muscle shows us that despite performing exercises aimed at preparing the hamstrings (Nordic hamstrings, for example), the semitendinosus remains more activated, probably most likely to protect the biceps femoris. This finding, on the one hand, may make us think that it is necessary to carry out more injury reduction work throughout the course of a season and, on the other hand, it demonstrates the suitability of fMRI to measure muscular activation in football players. Therefore, we can justify performing an injury reduction program to support rehabilitation in general as well as specific strengthening of the hamstrings, above all semitendinosus and biceps femoris. The program should be applied as a normal part of training as well as potencial application before the matches, to ensure that the hamstring muscle group is efficiently coordinated and working in harmony.

Another clinical implication associated with the results obtained is that if a player has an unusual injury pattern, we should include fMRI in the diagnostic approach to detect anomalies in the type of muscle activation during a football match. This can inform if a change of position on the field or a specific type of training may be necessary.

Finally, we may also use the information obtained from the fMRI to manage training loads during the week within professional football clubs.

## Acknowledgments

The authors sincerely thank to all FCBarcelona players and staff at the moment to collect the data.

## Author Contributions

**Conceptualization:** Sandra Mechó, Alicia Palomar-Garcia, Gil Rodas.

**Data curation:** Sandra Mechó, Alicia Palomar-Garcia, Gil Rodas.

**Formal analysis:** Juan R. González.

**Investigation:** Sandra Mechó, Alicia Palomar-Garcia, Gil Rodas.

**Methodology:** Sandra Mechó, Alicia Palomar-Garcia, Juan C. Gallego, Francesc López, Juan R. González, Gil Rodas.

**Project administration:** Manuel Wong, Xavier Valle, Ferran Ruperez, Ricard Pruna.

**Resources:** Manuel Wong, Xavier Valle, Ferran Ruperez, Ricard Pruna.

**Supervision:** Sandra Mechó, Alicia Palomar-Garcia, Juan R. González, Gil Rodas.

**Validation:** Sandra Mechó, Alicia Palomar-Garcia, Gil Rodas.

**Visualization:** Sandra Mechó, Alicia Palomar-Garcia, Gil Rodas.

**Writing – original draft:** Sandra Mechó, Alicia Palomar-Garcia, Gil Rodas.

**Writing – review & editing:** Juan R. González.

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
