## [Decision Letter · Decision Letter 0]

4 Jun 2024

PONE-D-24-10909Characterization of acute effects of football competition on hamstring muscles by muscle functional MRI techniquesPLOS ONE

Dear Dr. Mechó Meca,

Thank you for submitting your manuscript to PLOS ONE. After careful consideration, we feel that it has merit but does not fully meet PLOS ONE’s publication criteria as it currently stands. Therefore, we invite you to submit a revised version of the manuscript that addresses the points raised during the review process.

**Reviewers found merit in your manuscript but suggested revision. Please read through carefully and make sure your references are appropriately cited. Also consider the suggested change to "injury reduction".**

We look forward to receiving your revised manuscript.

Kind regards,

Jeremy P Loenneke

Academic Editor

PLOS ONE

Journal Requirements:

https://www.ncbi.nlm.nih.gov/pmc/articles/PMC4267196/

https://journals.plos.org/plosone/article?id=10.1371%2Fjournal.pone.0234401

In your revision ensure you cite all your sources (including your own works), and quote or rephrase any duplicated text outside the methods section. Further consideration is dependent on these concerns being addressed.

4. We note that Figure 1 includes an image of a patient in the study. 

Reviewers' comments:

Reviewer's Responses to Questions

**Comments to the Author**

1. Is the manuscript technically sound, and do the data support the conclusions?

Reviewer #1: Yes

Reviewer #2: Yes

2. Has the statistical analysis been performed appropriately and rigorously? 

Reviewer #1: N/A

Reviewer #2: Yes

3. Have the authors made all data underlying the findings in their manuscript fully available?

Reviewer #1: Yes

Reviewer #2: Yes

4. Is the manuscript presented in an intelligible fashion and written in standard English?

Reviewer #1: Yes

Reviewer #2: Yes

5. Review Comments to the Author

**Reviewer #1: **Very interesting study. The authors show that the semitendinosus is the most activated hamstring muscle during a soccer match and specifically in its middle and distal portion.

Abstract : Syntax errors « ). »

Introduction : a bit too long.

Materials and methods :

- « three portions for each, but in case of short head of biceps just two portions: proximal and distal). An average of 5 slices were discarded due to motion artifacts » : a limit of the study that has to be discussed.

- The MRI primarily represents the last minutes played. Have all the players played 60 minutes straight without stopping? It's not clear.

Results

Ok

Discussion

- « three portions for each, but in case of short head of biceps just two portions: proximal and distal). An average of 5 slices were discarded due to motion artifacts » : a limit of the study that has to be discussed.

- The MRI primarily represents the last minutes played : to be more discussed.

Fig 1 and 2 : images quality has to be improved.

**Reviewer #2: **The article as written provides the reader with novel information and may be useful in the understanding of hamstring injuries in general and specific to football athletes. I would have liked to see some more explanation and expounding upon the GPS data as it relates to the study and the findings. It felt that this portion of data collection was auxiliary to the study and not well utilized. There were found to be some hard to read sentences/sections of the manuscript as it relates to wording and flow. Please check this as well as typing errors found in the discussion section. Please check your use of references. There may have been some references or potential citation that may have been missed in the introduction section. I feel like the clinical implications as covered could have been more thorough, and the authors could include potential future research ideas or objectives based on their findings. The clinical implications state that the findings support rehab or strengthening of the semitendinosus as a preventative method after a match, however this may be better achieved to perform prior to the next match but not immediately after a match. Also, many believe that injury prevention is not a relative term and the term injury reduction is better suited. Overall, the article was well done, and provides a novel and interesting perspective.

6. PLOS authors have the option to publish the peer review history of their article (what does this mean?). If published, this will include your full peer review and any attached files.

Reviewer #1: **Yes: **Sylvain Grange, MD, PhD

Reviewer #2: No

---

## [Author Response · Author response to Decision Letter 0]

7 Jul 2024

Thanks for the time and effort taken in evaluating our manuscript and giving us useful suggestions. We sincerely think that the article has improved thanks to the reviewers instructions.

We have changed several paragraphs following the reviewers recommendations. All of them are indicated in the Response to reviewers document.

---

## [Decision Letter · Decision Letter 1]

23 Jul 2024

Characterization of acute effects of football competition on hamstring muscles by muscle functional MRI techniques

PONE-D-24-10909R1

Dear Dr. Mechó Meca,

We’re pleased to inform you that your manuscript has been judged scientifically suitable for publication and will be formally accepted for publication once it meets all outstanding technical requirements.

Kind regards,

Jeremy P Loenneke

Academic Editor

PLOS ONE

Additional Editor Comments (optional):

Reviewers' comments:

Reviewer's Responses to Questions

**Comments to the Author**

1. If the authors have adequately addressed your comments raised in a previous round of review and you feel that this manuscript is now acceptable for publication, you may indicate that here to bypass the “Comments to the Author” section, enter your conflict of interest statement in the “Confidential to Editor” section, and submit your "Accept" recommendation.

Reviewer #1: All comments have been addressed

2. Is the manuscript technically sound, and do the data support the conclusions?

Reviewer #1: Yes

3. Has the statistical analysis been performed appropriately and rigorously? 

Reviewer #1: Yes

4. Have the authors made all data underlying the findings in their manuscript fully available?

Reviewer #1: Yes

5. Is the manuscript presented in an intelligible fashion and written in standard English?

Reviewer #1: Yes

6. Review Comments to the Author

Reviewer #1: The questions have been answered correctly. The manuscript is intelligible and well written. The english is correct

7. PLOS authors have the option to publish the peer review history of their article (what does this mean?). If published, this will include your full peer review and any attached files.

Reviewer #1: No

---

## [Editor Report · Acceptance letter]

16 Aug 2024

PONE-D-24-10909R1 

PLOS ONE

Dear Dr. Mechó Meca, 

I'm pleased to inform you that your manuscript has been deemed suitable for publication in PLOS ONE. Congratulations! Your manuscript is now being handed over to our production team.

Kind regards, 

on behalf of

Dr. Jeremy P Loenneke 

Academic Editor

PLOS ONE